# Multidimensional Demographic Analyses of COVID-19 Vaccine Inequality in the United States: A Systematic Review

**DOI:** 10.3390/healthcare13020139

**Published:** 2025-01-13

**Authors:** Seyed M. Karimi, Sirajum Munira Khan, Mana Moghadami, Md Yasin Ali Parh, Shaminul H. Shakib, Hamid Zarei, Sepideh Poursafargholi, Bert B. Little

**Affiliations:** 1Department of Health Management and Systems Sciences, School of Public Health and Information Sciences, University of Louisville, Louisville, KY 40202, USA; seyed.karimi@louisville.edu (S.M.K.); mana.moghadami@louisville.edu (M.M.); shaminul.shakib@louisville.edu (S.H.S.); hamid.zarei@louisville.edu (H.Z.); 2Department of Bioinformatics and Biostatistics, School of Public Health and Information Sciences, University of Louisville, Louisville, KY 40202, USA; sirajummunira.khan@louisville.edu (S.M.K.); mdyasinali.parh@louisville.edu (M.Y.A.P.); sepideh.poursafargholi@louisville.edu (S.P.)

**Keywords:** COVID-19 vaccination, inequality, age, race, ethnicity, sex

## Abstract

Background: COVID-19 vaccination uptake is associated with demographic characteristics such as age, sex, and ethnicity-race in the United States (U.S.). Prior research predominantly analyzed COVID-19 vaccination uptake unidimensionally, limiting insights into multidimensional demographic inequalities. Multidimensional studies provide a closer insight into vaccination inequality and assist in designing more effective vaccination strategies. Objectives: Review descriptive studies of the COVID-19 vaccination uptake across combinations of at least two of the three key demographic characteristics: age, sex, and ethnicity-race in the U.S. Methods: A systematic review was performed using the Joanna Briggs Institute methodology and adhering to the PRISMA-ScR principles for reporting. Six impartial reviewers examined all of the papers. The data were obtained using a tailored data extraction template. Results: A total of 2793 records were initially downloaded, 461 of them were dropped for duplication, and 2332 were reviewed. Based on the title and abstract reviews, 2115 records were excluded. After reviewing the full text of the remaining records, 212 more records were excluded. The remaining six records were reviewed to identify and compare their population, study period, data, the studied dose number, methodology, and results. Conclusions: Multidimensional COVID-19 vaccine uptake analyses are rare and mostly focused on the dose-one vaccination. Improving researchers’ access to immunization registry data while preserving data security is a prerequisite for such analyses.

## 1. Introduction

This study reviewed evidence of inequalities in the COVID-19 vaccine uptake by combinations of three commonly reported demographic characteristics (age, sex, and ethnicity-race) in the United States (U.S.). This study focused on the U.S. to preserve homogeneity in the national immunization policies and the documentation of race and ethnicity.

The COVID-19 pandemic exposed the vulnerabilities of the U.S. healthcare systems and disparities in health and healthcare access in different demographic groups. Inequality in COVID-19 outcomes (especially death and hospitalization) across sex, age, racial, and ethnic groups has been documented in the U.S. Sex and age differences in COVID-19 mortality rate were persistent, such that the rate was constantly higher in the males compared to females and in older individuals than younger. For example, in January 2021, amid the largest surge in COVID-19 cases in the U.S., the death rate per 100,000 individuals was 28.0 in females but 35.5 in males [1]. Also, in January 2021, the COVID-19 mortality rate among 75-plus, 65–74, 50–64, 40–49, 30–39, and 18–29-year-old Americans was 275.9, 71.4, 24.8, 6.9, 2.4, and 0.7 per 100,000, respectively [1].

Racial and ethnic differences in COVID-19 mortality were not persistent over time in the U.S. For instance, the highest COVID-19 mortality rate among racial and ethnic groups was reported among non-Hispanic Black individuals by May 2020 and then among non-Hispanic American Indian or Alaska Native (AI/AN) individuals from June 2020 to February 2021 [1]. The COVID-19 mortality rate for non-Hispanic White individuals was one of the lowest among racial and ethnic groups by September 2020, but it became one of the highest rates from August 2020 to March 2022 and remained the highest rate thereafter [1]. Conversely, the COVID-19 mortality rate among Hispanic individuals was one of the highest by August 2020 but one of the lowest thereafter [1]. The most persistent pattern of COVID-19 mortality rate was recorded among non-Hispanic Asian individuals, who had the lowest rate among racial and ethnic groups since the beginning of the pandemic [1].

Age-specific COVID-19-related hospitalization rates illustrate similar nationwide trends to the COVID-19 mortality rates in the U.S. The rate of COVID-19-related hospitalization was consistently higher among older individuals [2,3,4]. On the other hand, males and females have taken turns having a higher hospitalization rate due to COVID-19 [5]. Among races and ethnicities, non-Hispanic Black and non-Hispanic AI/AN individuals usually had the highest, and non-Hispanic Asian and Pacific Islander individuals had the lowest hospitalization rates [2,3,4].

Variations in mortality and hospitalization because of COVID-19 among demographic groups in the U.S. are attributed to disparities in vaccine uptake, among other factors [6,7,8]. Temporal data regarding COVID-19 vaccination rates by demographics is far less abundant than data on COVID-19-related mortality and hospitalization. Nonetheless, limited evidence indicates that COVID-19 immunization trends align with the trends in COVID-19 outcomes. A study conducted shortly after the start of the immunization campaign in the U.S. used states’ immunization registry data collected by the Centers for Disease Control and Prevention (CDC) and showed that 58.0% of adult female Americans received at least one dose of the COVID-19 vaccine by 22 May 2020; the rate for adult male Americans was 53.4% [9]. By May 2023, the dose-one vaccination rate among females of all ages reached 82%, while it reached 77% in males, maintaining the approximately 5% sex gap [10]. On 22 May 2021, dose-one COVID-19 vaccination inequality by age group was remarkable: 79.1% in 65-plus, 62.9% in 50–64, 48.5% in 30–49, and 37.6% in 18–29 year old adults [9]. Similar results were found in another study that used data between April 2021 and November 2021 from the CDC’s National Immunization Survey Adult COVID Module (NIS-ACM) [11].

The NIS-ACM study also showed that the national-level dose-one COVID-19 vaccination rate was persistently higher among non-Hispanic Asian adults (18 and older) than among other races and ethnicities for most of the December 2020 to November 2021 period [11]. The second highest rate was reported among non-Hispanic White adults until August 2021, followed by a trend shift to Hispanic adults from September 2021 to November 2021. On the other hand, the rate was continuously lower than the average among non-Hispanic Black adults and the lowest among non-Hispanic AI/AN adults. In April 2021, for example, 69.6% of non-Hispanic Asian adults received at least one dose of a COVID-19 vaccine. The rate for non-Hispanic White, Hispanic, non-Hispanic Black, and non-Hispanic AI/AN adults was 47.3%, 59.0%, 46.3%, and 38.7%, respectively [11]. In November 2021, the rate for non-Hispanic Asian adults increased to 95.2%, and for non-Hispanic White, Hispanic, non-Hispanic Black, and non-Hispanic AI/AN adults to 78.7%, 81.3%, 78.2%, and 61.8%, respectively [11]. A Kaiser Family Foundation report, using vaccination data by race and ethnicity from 36 U.S. states during March 2021–July 2022, found largely similar results: non-Hispanic Asian individuals of all ages had the highest dose-one vaccination uptake from May 2021 to July 2022, reaching 87% by July 2022, compared to 67%, 64%, and 59% in Hispanic, non-Hispanic White, and non-Hispanic Asian individuals, respectively [12].

Unidimensional demographic evidence on COVID-19 outcomes and immunization rates aids in recognizing disparities among single-factor demographic groups but lacks specificity for formulating effective public health protection strategies. Multidimensional demographic evidence (e.g., disparities in the COVID-19 vaccination rate by races and ethnicities within specific age, sex, or age–sex groups) allows for more accurate identification of inequalities in the population, establishing priority groups and informing the development of effective public health interventions.

Multidimensional demographic evidence on COVID-19 outcomes in the U.S. is not scarce. For instance, the CDC maintains a data dashboard that illustrates temporal patterns in COVID-19 mortality by age within racial and ethnic groups and by race and ethnicity within age groups [1]. The CDC also maintains a data dashboard that provides information on COVID-19 hospitalization rates by age within racial and ethnic groups and by age within sex groups. However, the CDC does not provide such evidence on COVID-19 vaccination uptakes. Moreover, the literature on the multidimensional demographic disparity in COVID-19 vaccination uptake is limited.

This systematic review was designed to collect and analyze published records of descriptive multidimensional demographic evidence on COVID-19 vaccination uptake, measured by COVID-19 vaccination rate. This study’s research question was: What were the inequalities in the COVID-19 vaccination rate by the composition of at least two demographic characteristics among age, sex, race, and ethnicity in the U.S.? Specifically, this systematic review consisted of studies that reported differences in COVID-19 vaccination rates among racial and ethnic groups within different age groups by sexes (e.g., between males and females among 5–11-year-old Asian individuals).

## 2. Materials and Methods

### 2.1. Eligibility Criteria

This study was designed to collect evidence on the COVID-19 vaccination inequalities by combinations of at least two of the three key demographic characteristics—namely, age, sex, and ethnicity-race—in the U.S. These demographic characteristics are commonly reported with minimal variations, especially in race and ethnicity, in U.S. data such as decennial censuses, immunization registries, and population, community, and household surveys. However, U.S. data do not commonly report socioeconomic characteristics such as education, marital status, and family income. Notably, U.S. immunization registries do not contain such information. Ethnicity and race were combined and notated as “ethnicity-race”, as they are usually reported as Hispanic (including Hispanic Asian, Hispanic Black, and Hispanic White) versus non-Hispanic (including non-Hispanic Asian, non-Hispanic Black, and non-Hispanic White) in the U.S. publications. COVID-19 vaccination inequality was measured by the differences in the rates of vaccinated individuals in the demographic groups.

### 2.2. Search Strategy and Information Sources

Six researchers reviewed the related literature in 24 months. A systematic review methodology was used for this study because of the lack of published reviews (including literature, rapid, scoping, umbrella, or systematic review) on the topic, the assessment of the risk of bias, and the synthesis of the included studies’ findings [13,14]. The Preferred Reporting Items for Systematic Reviews and Meta-Analyses (PRISMA) reporting guidelines were followed [15]. A protocol was not registered for this review.

Search keywords were established on the following concepts: location, COVID-19, vaccine, uptake, demographic factors, and inequality (Table 1). Databases Embase, ProQuest, PubMed, and Web of Science (WOS) were searched for the related articles from 1 January 2020 to 31 December 2023. Using the keywords, database-specific search strings were designed and applied (Appendix A). The reference lists of the included articles were manually examined to capture any relevant studies that may not have been included in the initial search strategy.

### 2.3. Study Selection

All articles that met the following criteria were included in the systematic review: (1) originally written in English, (2) examined U.S. data, (3) assessed COVID-19 vaccine administration, injection, or uptake, (4) focused on disparities in vaccine uptake in two or more combinations of age, sex, and ethnicity-race, and (5) provided direct descriptive evidence on vaccination rates. Studies that only focused on the distribution or rollout of the COVID-19 vaccine, focused on COVID-19 vaccine hesitancy, analyzed COVID-19 vaccine inequality by a single demographic factor, or examined multidimensional demographic disparity in COVID-19 vaccination uptake only through regression analyses were excluded.

### 2.4. Data Collection Process

All articles collected from the separate database searches were imported into Endnote version 21. In the next step, the duplicated articles were deleted. Then, the articles’ titles and abstracts were screened by two independent reviewers (Sirajum Munira Khan and Mana Moghadami). A third reviewer (Seyed M. Karimi) was consulted on the articles that were not included by both of them. The remaining articles’ full text was reviewed by three independent reviewers (Sirajum Munira Khan, Mana Moghadami, and Seyed M. Karimi) and discussed to identify studies that met all the inclusion criteria. Those studies were included in this review. The process was depicted in a PRISMA flow chart [16].

### 2.5. Data Items

After identification of the eligible articles, a data extraction table was made with categories containing year, first author, studied COVID-19 doses, study population information (if focused on the general population or a specific sub-population, location, and age range), study period (year, month, and time from the start of the COVID-19 vaccination campaign in the U.S. in December 2020), data (data type, dataset name, observational unit, and sample size), methods (analysis type, results presentation, combined demographic subclassifications, and studied age, sex, race, and ethnic groups), and key findings. All data types, including surveys and immunization registry data, were considered.

### 2.6. Reporting Bias Assessment

All the selected articles were assessed for quality assessment according to the checklist for assessing the quality of quantitative studies in Standard Quality Assessment Criteria [17]. Three authors (Sirajum Munira Khan, Mana Moghadami, and Seyed M. Karimi) independently assessed the quality of each study.

### 2.7. Effect Measures

The presentation of findings (i.e., COVID-19 vaccination rates) at points in time, time trends, and maps was reviewed. Also, the demographic multidimensionality of the findings was inspected. Considering ethnicity-race as one demographic dimension, demographic characteristics could have been combined in pairs of two or three: age–sex, age–ethnicity-race, sex–ethnicity-race, and age–sex-ethnicity-race. Three authors (Sirajum Munira Khan, Mana Moghadami, and Seyed M. Karimi) filled out the data extraction table independently and then discussed them.

### 2.8. Synthesis Method and Subgroup Analyses

The included articles’ studied populations in terms of location and age range, study periods, data sources, units of observations, and sample sizes were reviewed and juxtaposed. Their methods were also assessed in terms of analysis type, results presentation, demographic multidimensionality, specificities of age, sex, and ethnic-racial groups, and quality. Their findings were described by dose number, demographic multidimensionality, and large age groups (namely, older adults, adults, and children). Specifically, the findings for each dose number were reviewed separately. The results were compared by demographic combination and then large age group within each dose. A meta-analysis could not be conducted because of differences among included studies in the studied COVID-19 dose number, age and ethnicity-race subcategories, and the study period.

## 3. Results

### 3.1. Study Selection

Applying the search strings to the databases led to identifying 2793 documents: 1017 from Embase, 454 from ProQuest, 169 from PubMed, and 1153 from WOS (Figure 1). Combining the documents from the four sources, followed by eliminating the duplicate reports (461 records), resulted in 2332 unique documents. Among them, 2115 were excluded based on title and abstract reviews. Therefore, 217 studies were included for full-text review, which led to the exclusion of 211 documents that did not meet a key inclusion criterion: providing evidence on the COVID-19 vaccine uptake disparities in two or more combinations of age, sex, and ethnicity-race. Among the six included documents, one that examined multidimensional COVID-19 vaccine uptake only through regression analyses was excluded as it provided indirect, covariate-adjusted estimates of multidimensional COVID-19 vaccine uptake inequality. Ultimately, the remaining five peer-reviewed articles, plus one found after snowballing references, were assessed as eligible for inclusion in this systematic review (Figure 1).

### 3.2. Study Characteristics: Population, Period, Data, and the Dose Numbers

All the qualified articles were built on individual-level vaccination data (Table 2). In three studies, the data were collected through surveys [11,18,19]; the other three studies used population-level data from immunization registries [9,20,21]. Five of the six were national studies [9,11,18,19,20], and one focused on a specific city, New York City, NYC [21].

The selected articles primarily studied adults aged 18 years and older (Table 2). However, two articles exclusively focused on children aged 5 or older [19,21]. One article included 5–17-year-old children in addition to 18-plus-year-old adults [20]. The subcategories of children’s age accorded with the timeline of the Food and Drug Administration’s (FDA) emergency use authorization (EUA) for Pfizer and Moderna vaccines: 12–15, 5–11, and 0.5–4 years with EUA dates of 10 May 2021, 29 October 2021, and 17 June 2022, respectively. Children aged 16 years or older were considered a separate age group because the first Pfizer vaccine was permitted for them on its initial EUA date, 11 December 2020 [10,19,21,22,23]. Adult age groups, however, varied from one study to another [19,21].

One-dose COVID-19 vaccine recipients were studied more frequently in multidimensional studies compared to two or more doses of COVID-19 vaccine recipients (Table 2): three of the six studies analyzed only dose-one vaccination [9,11,18], one study focused only on dose-two vaccination [21], another study examined doses one, two, and three [19], and the other assessed doses three and four [20]. The sample size varied significantly across the selected studies: 321 [18] to 255,200,373 [9].

### 3.3. Study Characteristics: Methods

Two studies provided the evidence only at cross-sections of time [18,20]; three provided both cross-sectional tables and time trends [9,11,19], and one provided only time trends [21] (Table 3). The largest time period covered by a study was the first 21 months of COVID-19 vaccination, from December 2020 to August 2022 [20]. Neither of the included studies provided evidence of geographical disparities in COVID-19 vaccination. Regarding the risk of bias, the quality of five studies was assessed Very Good [9,11,19,20,21], one Good [18].

The multidimensionality of the studies was limited to two or three dimensions: the comparison of sexes within age groups (i.e., age–sex combinations), the comparison of ethnicity-race combinations within age groups (i.e., age–ethnicity-race combinations), and the comparison of ethnicity-race combinations within sex groups (i.e., sex–ethnicity-race combinations) (Table 3). All studies defined Hispanic ethnicity mutually exclusively from racial groups. That is, the racial and ethnic groups were defined as Hispanic, non-Hispanic White, non-Hispanic Black, non-Hispanic Asian, etc. As a result, Hispanic White, Hispanic Black, and Hispanic Asian individuals were not distinguished. Accordingly, the phrase “ethnicity-race”, which indicates races within ethnic groups to describe this specific aspect of multidimensionality, was used in this study.

Age–sex combinations, for example, female versus male 65-plus-year-olds or female versus male 40–64-year-olds, were analyzed in two studies [9,20] (Table 3). Two studies analyzed sex–ethnicity-race combinations, for example, Hispanic versus non-Hispanic Black females [18,19]. Except for study [9], the other studies analyzed age–ethnicity-race combinations, for example, 5–11-year-old non-Hispanic Black versus 5–11-year-old Hispanic individuals or 12–15-year-old non-Hispanic Black versus 12–15-year-old Hispanic individuals [11,18,19,20,21]. Notably, none of the studies analyzed age–sex combinations within racial or ethnic groups, for example, female 5–11-year-old non-Hispanic White versus female 5–11-year-old non-Hispanic Black individuals.

### 3.4. Results

The findings of the included studies could not be statistically synthesized due to dissimilarities in the analyzed dose, study period, and categories of the subcategories of demographic characteristics. In the following, their findings are described by dose number, demographic combination, and large age groups.

#### 3.4.1. Results from Dose-One Studies

##### Analyses of Age–Ethnicity-Race Combinations

Three studies provided evidence of the dose-one COVID-19 vaccine disparity by a number of age–ethnicity-race combinations [11,18,19]. Two focused on adults [11,18], and one on children [19].


*Older Adults*


One adults study combined all older adults—individuals 65 years or older—in one group [11], whereas the other study used the 60–69, 70–79, and 80–88-year age groups [18]. The first used the CDC’s NIS-ACM and reported an 82.8% dose-one vaccination rate among non-Hispanic White older adults in April 2021—five months after the start of mass vaccination in the U.S. [11]. The rate was higher in non-Hispanic Asian and Native Hawaiian/Other Pacific Islander (NH/OPI) older adults: 85.5% and 83.0%, respectively, but lower in Hispanic, non-Hispanic Black, non-Hispanic Multiracial or Other Races (M/OR), and AI/AN older adults: 75.4%, 74.3%, 69.9%, and 60.9%, respectively [11]. Reporting the same in November 2021, all rates increased, and disparities showed a decrease in most cases. The rate among non-Hispanic White older adults was 94.1%: 95.3%, 97.6%, and 99.6% among Hispanic, non-Hispanic Asian, and NH/OPI older adults, respectively; 92.9%, 91.2%, and 85.7% among non-Hispanic Black, M/OR, and AI/AN people, respectively [11].

The other adults study used a small sample from the Household Pulse Survey (HPS) conducted from July to October 2021—8 to 11 months after the vaccination campaign—and reported the highest vaccination rate at ages 70–79 and 60–69 years for non-Hispanic Asian older adults: 92.8% and 91.1%, respectively [18]. At ages 80–88, however, the rate was 69.6% among non-Hispanic Asians versus 84.4% among non-Hispanic White older adults and 89.1% among non-Hispanic Black older adults [18]. In all three age groups, the lowest rates were reported for Hispanic individuals: 41.1%, 84.4%, and 85.3% at ages 80–88, 70–79, and 60–69, respectively [18].


*Adults*


One adults study divided younger adults—individuals between 18 and 64—into three age groups: 18–29, 30–49, and 50–64 years [11], while the other divided them into four groups: 18–29, 30–39, 40–49, and 50–59 years [18]. For adults aged 18 to 29, the first study reported the highest vaccination rate, 60.0%, in non-Hispanic Asian individuals and the lowest rate, 21.0%, in NH/OPI individuals in April 2021 [11]. The second used using July–October 2021 data and found the highest dose-one vaccination rate in non-Hispanic Asian individuals, 89.3%, and the lowest rate, 48.7%, in non-Hispanic Black individuals, although it did not include AI/AN, NH/OPI, and M/OR individuals in the analysis [18]. In November 2021, the first study again reported that non-Hispanic Asian individuals had the highest vaccination rate, 92.7%, but the lowest vaccination rate, 45.2%, this time belonged to AI/AN individuals [11].

In April 2021, the highest vaccination rates at ages 30–49 and 50–64 years were reported for non-Hispanic Asian individuals: 70.4% and 77.6%, respectively; the lowest rates at these ages were reported for AI/AN individuals at 34.6% and 43.1%, respectively [11]. By November 2021, the dose-one vaccination rate among individuals of all ethnicity-race groups at these ages significantly increased, but the highest vaccination rates were still in non-Hispanic Asian individuals at 95.5% and 98.1%, respectively, the lowest vaccination rates in AI/AN individuals at 55.5% and 68.3%, respectively [11].

In July–October 2021, the highest dose-one vaccination rate in 30–39, 40–49, and 50–59-year-old individuals pertained to non-Hispanic Asian individuals: 89.6%, 91.6%, and 92.9%, respectively [18]. However, the lowest rates at ages 30–39 and 40–49 were 54.7% and 70.7%, respectively, and belonged to non-Hispanic Black individuals; at ages 50–59, Hispanic individuals had the lowest rate at 80.4% [18].


*Children*


A study using the CDC’s NIS-CCM from July 2021 to September 2022 divided children into three groups: 5–11, 12–15, and 16–17 years [19]. In all these age groups, the highest rates were reported for non-Hispanic Asian children: 63.5%, 87.3%, and 87.3% at ages 5–11, 12–15, and 16–17, respectively [19]. At ages 5–11, 12–15, and 16–17, the lowest dose-one vaccination rate belonged to non-Hispanic White children, at 32.0%, 54.5%, and 65.8%, respectively [19]. The study did not include AI/AN, NH/OPI, and M/OR individuals in their analysis.

##### Analyses of Sex–Ethnicity-Race Combinations

Two studies examined dose-one COVID-19 vaccine disparity by a set of sex–ethnicity-race combinations [18,19]. The studies focused on two mutually exclusive age groups: adults 18 years or older [18] and children under 18 [19].


*Adults*


The adults study reported the highest vaccination rates in both female and male non-Hispanic Asian individuals in July–October 2021: 90.8% and 90.3%, respectively, and the lowest rates in female and male non-Hispanic Black individuals: 70.0% and 71.8%, respectively [18]. The study also reported a slightly higher female than male vaccination rate in all ethnicity-race groups, except in the non-Hispanic Black group [18].


*Children*


The children study measured a largely small difference in vaccination rate by sex among children under 18 in August 2022 [19]. The female-male difference in dose-one vaccination rate was the highest in non-Hispanic White children, 2.8%, but the lowest in −0.2% in non-Hispanic Black children [19].

##### Analyses of Age–Sex Combinations

One study reported the dose-one COVID-19 vaccination rate among adult age groups by sex in May 2021—about six months after the country’s COVID-19 vaccination program was launched [9]. They reported a lower vaccination rate in males than females among preretirement adults: 3.5%, 4.7%, and 6.4% lower in 50–64, 30–49, and 18–29 year-olds; but a higher vaccination rate in males than females among older adults: 2.4% higher. Overall, the vaccination rate among adult males was 4.6% lower than adult females [9].

#### 3.4.2. Results from Dose Two Studies

##### Analyses of Age–Ethnicity-Race Combinations

Evidence on the dose two COVID-19 vaccine disparity by a set of age–ethnicity-race combinations was provided in two studies: both focused on children, one on children under 18 in a national study using CDC’s NIS-CCM data [19] and the other on under 19 NYC public school children [21]. The analysis of age groups 5–11 and 12–15 was common between both studies. The NYC study presented results for 180 days after each age group became eligible for COVID-19 vaccination: 13 May 2021 to 28 October 2021 for 12–15-year-olds and 4 November 2021 to 2 May 2022 for 5–11-year-olds [21,22]. The national study measured children’s vaccination rates from July to September 2022 [19].

An approximately 85%, 62%, 61%, 55%, and 44% (percentages inferred from a trends figure) two-dose vaccination rate was reported, respectively, in non-Hispanic Asian, non-Hispanic M/OR, Hispanic, non-Hispanic White, and non-Hispanic Black 12–15-year-old children on 28 October 2021 in NYC public schools [21]. Nine to eleven months later, the national data collected from 1 July to 30 September 2022 showed 86.6%, 57.5%, 57.8%, 52.2%, and 53.1% two-dose vaccination rates in non-Hispanic Asian, non-Hispanic OR, Hispanic, non-Hispanic White, and non-Hispanic Black 12–15-year-old children, respectively [19].

Both studies showed a large difference in the vaccination rate of 5–11-year-old children compared to 12–15-year-old children. The NYC study reported an approximately 73%, 50%, 40%, 42%, and 33% (percentages inferred from a trends figure) two-dose vaccination rate, respectively, in non-Hispanic Asian, non-Hispanic OR, Hispanic, non-Hispanic White, and non-Hispanic Black 5–11-year-old children on 2 May 2022 in New York City public schools [21]. Three to five months later, the national study reported 57.1%, 28.8%, 28.8%, 29.0%, and 27.3% vaccination rates, respectively, in non-Hispanic Asian, non-Hispanic M/OR, Hispanic, non-Hispanic White, and non-Hispanic Black 5–11-year-old children [19].

#### 3.4.3. Results from Booster Dose Studies

##### Analyses of Age–Ethnicity-Race Combinations

Evidence on the COVID-19 first booster dose vaccine disparity by a set of age–ethnicity-race combinations was provided in two studies [19,20]. One focused on children under 18 [19], but the other included all age groups [20]. These studies provided a snapshot of the booster dose vaccination rate in the summer of 2022: August 2022 using immunization registry data [20] and July–September 2022 using the CDC’s NIS-CCM survey [19]. Both studies reported the highest first booster dose vaccination rates among non-Hispanic Asian individuals in all studied age groups [19,20]. The lowest rate was reported in non-Hispanic Black individuals in all age groups younger than 40 years [19,20]. One reported the lowest rate at ages 40–64 and 65-plus in Hispanic individuals [20].

##### Analyses of Age–Sex Combinations

One article studied the administration of booster doses among both sexes across children, adults, and older adults [20]. For the first and second booster doses, its findings indicated that women had higher vaccination rates across all age groups, with the exception of the 5–11 age group, where vaccination rates were equal between girls and boys, both at 15.6% [20].

## 4. Discussion

### 4.1. A Summary of the Key Study Characteristics

Most multidimensional demographic studies of the COVID-19 vaccine uptake in the U.S. used survey data, which can potentially lead to sample size and potential selection bias issues (Table 2). One study’s problems were more concerning, as its sample size was very small [18]. In addition to the concern about the generalizability of the results of these studies, their sample sizes did not allow for identifying many racial-ethnic groups. Such issues are addressed when immunization registry data are used [9,19,20]. Nonetheless, immunization registries were most commonly used at the national level [9,20]. Therefore, geographical disparities in COVID-19 vaccine uptake by a combination of demographic characteristics (for example, in non-Hispanic White versus non-Hispanic Black 12–15-year-old children) are largely unknown. In addition, most multidimensional studies focus on receiving one dose of a COVID-19 vaccine; hence, evidence on the uptake of two or more doses of the COVID-19 vaccine is scarce.

Most studies provided multidimensional evidence of differences in COVID-19 vaccine uptake at a cross-section of time (Table 3). Time trends, which indicate the pattern of divergence or convergence in vaccination rate across demographic characteristics—hence, identifying groups that might have delayed access to the vaccine—are less frequently used. Additionally, no study that geographically analyzed multidimensional differences in COVID-19 vaccination rate was found.

Age–ethnicity-race was the most frequently analyzed combination of demographics regarding the COVID-19 vaccine uptake, followed by sex–ethnicity-race and age–sex (Table 3). Importantly, neither of the studies provided a multidimensional analysis of dose-two vaccine uptake, indicating the completion of the COVID-19 vaccination series—also called full vaccination—in adults. Moreover, no study was found to conduct a triple-level analysis, considering age–sex-ethnicity-race combinations, for example, to compare COVID-19 vaccination uptake in non-Hispanic White versus non-Hispanic Black 12–15-year-old female children.

### 4.2. Key Findings, Relevance to the Literature, and Policy Implications

The review of the results of the selected articles revealed some consistent patterns in COVID-19 vaccination. First, gaps in COVID-19 vaccination decreased over time across age and race-ethnic groups. For example, in August 2022, about two years after the start of COVID-19 vaccination, gaps in booster dose vaccination rate were much smaller than dose-one vaccination years in the first year of vaccination [11,19,20].

Second, non-Hispanic Asian individuals consistently had the highest rate of COVID-19 vaccination regardless of age, sex, and dose number. Also, the gap in COVID-19 vaccination between non-Hispanic Asian individuals and the rest was inversely related to age: the younger, the larger the gap. The high rates of COVID-19 vaccination among non-Hispanic Asian individuals can be at least partially attributed to low vaccine hesitancy among them in comparison to those of other races and ethnicities [24,25,26]. This promising outcome, however, should be interpreted carefully as all non-Hispanic Asian individuals are combined as one group in the reviewed studies. Major non-Hispanic Asian groups in the U.S. include those of Chinese, Indian, Vietnamese, Korean, Japanese, and other Asians, but they differ significantly in terms of economic indicators [27]. Since economic well-being has been shown to be strongly correlated with COVID-19 vaccine intent and uptake [24,28], disparities in COVID-19 vaccination among non-Hispanic Asian Americans are expected. This necessitates the need to generate data and conduct research with more detailed racial information to identify vulnerable groups within the non-Hispanic Asians in the U.S.

Third, non-Hispanic White adults had the second highest dose-one vaccination rate across most adult age groups by two-quarters after the start of the COVID-19 vaccination campaign in December 2020. From then on, the dose-one vaccination rate in non-Hispanic White adults fell below that in Hispanic adults. Also, one of the lowest vaccination rates was found in non-Hispanic White children [19,21]. More detailed studies need to be conducted to analyze the low COVID-19 vaccination rate in non-Hispanic White children to understand its variation by sex, neighborhood, and family income to develop appropriate messaging strategies and interventions.

Fourth, Hispanic adults had the lowest dose-one vaccination rate in the first couple of quarters of the COVID-19 vaccination campaign in the U.S. by a large margin compared to non-Hispanic White adults [11]. However, the gap was filled rapidly in the following quarters, such that they had the second-highest dose-one vaccination rate by the end of 2021 [11]. The lag indicates the importance of timely outreach to the Hispanic community at the time of public health and addressing potential barriers to access to care, for example, English level, higher education, and access to health insurance [29].

Fifth, alongside Hispanic individuals, non-Hispanic Black individuals had one of the lowest booster dose vaccination rates [20]. An exception was the non-Hispanic Black older adults’ booster dose vaccination rate. Moreover, the difference in the uptake of dose-one vaccine between non-Hispanic Black older adults versus older adults in other racial-ethnic groups was small. The inverse relationship between COVID-19 vaccine hesitancy and age among Black Americans may be partially responsible for the high COVID-19 vaccine uptake in non-Hispanic older adults but low uptake in younger adults [30]. Hence, COVID-19 vaccine informational programs among non-Hispanic Black younger adult may reduce the gap in their COVID-19 vaccine uptake.

Sixth, there is limited evidence on the COVID-19 vaccination by age and sex in AI/AN, NH/OPI, and M/OR individuals, usually due to small sample sizes. The results from the CDC’s NIS-ACM showed that NH/OPI 30-plus-year adults had one of the highest dose-one vaccination rates in the second and fourth quarters of vaccination, but AI/AN and M/OR lagged significantly behind [11]. Considering the scarcity of information on COVID-19 vaccine uptake in these racial minority groups by demographic details, data availability and research in these areas should be a public health priority.

### 4.3. Limitations

This systematic review has several limitations. First, it solely focuses on the U.S. Hence, its results are not generalizable to other regions and countries. Second, this study covers the first three years (2021–2023) of the COVID-19 vaccination campaign in the U.S.; hence it is possible that more multidimensional evidence on COVID-19 vaccine uptake was collected in 2024. Third, this study did not review multidimensional evidence on COVID-19 vaccine inequality by socioeconomic characteristics (e.g., education, marital status, and income), mainly because such information is not reported in U.S. immunization registries and at the individual level in census data. Therefore, reviewing the analyses of the intersection of demographic and socioeconomic factors remains an area of development in the COVID-19 vaccination literature. Fourth, it is possible that some articles used keywords other than what is listed in Table 1 to describe location, demographic characteristics, or vaccine inequality. In particular, some of the databases (namely, Web Of Science and ProQuest) had a limit to the number of searched keywords. Fifth, there may be local data dashboards that provided multidimensional evidence on COVID-19 vaccine inequality in the U.S., although no such dashboards were found at the national and state levels. Sixth, since age groups were not the same in the selected articles, comparing COVID-19 vaccination rates by exact similar ages by sex or ethnicity-race was not possible. Seventh, articles presenting only model-based results, such as outputs from logistic or linear regression models, that incorporated any combination of demographic characteristics as interaction covariates were excluded, as they did not allow for homogeneous comparison in terms of adjusted covariates and method [31,32,33].

## 5. Conclusions

This study is the first to systematically review peer-reviewed publications that analyzed multidimensional demographic evidence on COVID-19 vaccine uptake in the U.S. Such analyses often focused on the dose-one vaccination. Similar analyses of full vaccination (i.e., the receipt of the first two doses), especially in older adults, are rare but crucially important because of the marked age gradient of COVID-19 vulnerability.

Detailed multidimensional demographic analyses of immunization allow for more accurate identification of the under-vaccinated groups and for designing more optimal and cost-effective strategies to address COVID-19 vaccine disparities. Improving researchers’ access to immunization registry data while preserving data security is a prerequisite for such analyses.

## Figures and Tables

**Figure 1 healthcare-13-00139-f001:**
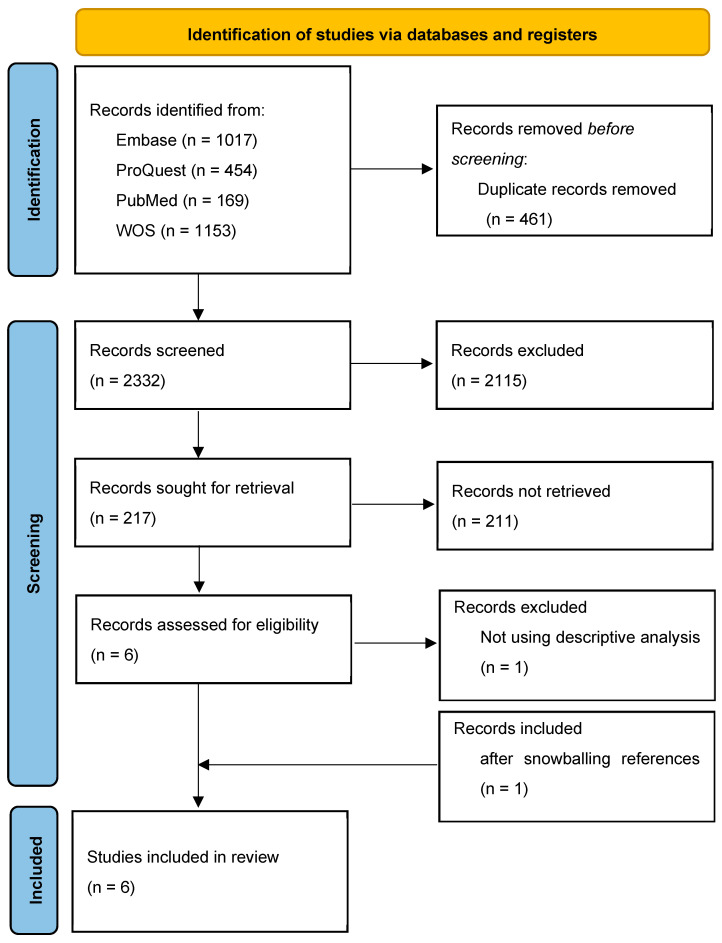
Study selection flow chart.

**Table 1 healthcare-13-00139-t001:** Search Keywords.

General Concepts	Keywords
Location	United States, U.S., US, U.S.A, USA, Alabama, AL, Alaska, AK, Arizona, AZ, Arkansas, California, CA, Colorado, CO, Connecticut, CT, Delaware, DE, District of Columbia, DC, Florida, FL, Georgia, GA, Hawaii, HI, Idaho, ID, Illinois, IL, Indiana, IN, Iowa, IA, Kansas, KS, Kentucky, KY, Louisiana, LA, Maine, ME, Maryland, MD, Massachusetts, MA, Michigan, MI, Minnesota, MN, Mississippi, MS, Missouri, MO, Montana, MT, Nebraska, NE, Nevada, NV, New Hampshire, NH, New Jersey, NJ, New Mexico, NM, New York, NY, North Carolina, NC, North Dakota, ND, Ohio, OH, Oklahoma, OK, Oregon, OR, Pennsylvania, PA, Rhode Island, RI, South Carolina, SC, South Dakota, SD, Tennessee, TN, Texas, TX, Utah, UT, Vermont, VT, Virginia, VA, Washington DC, Washington, WA, West Virginia, WV, Wisconsin, WI, Wyoming, WY
COVID-19	COVID, COVID19, COVID-19, Corona, Coronavirus, 2019-nCoV, SARS-CoV-2, Severe Acute Respiratory Syndrome Coronavirus 2
Vaccine	Vaccine, Vaccination, Vaccinated, Immunization, Immunisation, Injection, Injected
Uptake	Uptake, Administration, Receipt, Coverage, Rate, Rates, Service, Completion, First-Dose, First Dose, Dose-One, Dose One, Second-Dose, Second Dose, Dose Two, Partially-Vaccinated, Partially Vaccinated, Fully-Vaccinated, Fully Vaccinated, Booster, Boosted
Demographic Characteristics	Demographic, Sociodemographic, Socio-demographic, Socioeconomic, Socio-economic, Social Class, Race, Racial, Ethnicity, Ethnic, Race-Ethnicity, Ethnicity-Race, Racial-Ethnic, Racial/Ethnic, Black-White, Black/White, Black, Blacks, African American, African American Black, Hispanic, Latino, Latinos, Latinx, Hispanic/Latino, BIPOC, Indigenous, Sex, Gender, Age, Aged, Geriatric, Age-Group, Age Group, Elderly, Medicare, Child, Children, Adolescent, Pediatric
Inequality	Equity, Equality, Equitable, Inequity, Inequities, Inequality, Inequalities, Unequal, Disparity, Disparities, Healthcare Disparities, Discrimination, Community Vulnerability, Disadvantage, Disproportionate, Gap, Difference, Cluster, Clustering Factors

**Table 2 healthcare-13-00139-t002:** The Included Articles’ Population, Study Period, Data, and the Studied Dose Numbers.

		**Studied**	Population			Study Period					
#	Authors (Year)	Dose Numbers	Sub- Population	Location	Age Range	Duration	Time from the Start of Vaccination	Data Type	Data		
									Dataset Name	Obs. Unit	Sample Size
1	Diesel et al. (2021) [9]	1	General Population	U.S.	18+	December 2020 to May 2021	Months 1–6	Immunization Registry	States and the D.C. immunization Registries’ Data Submitted to the CDC Immunization Information Systems	Individual	255,200,373
2	Kriss et al. (2022) [11]	1	General Population	U.S.	18+	December 2020 to November 2021	Months 1–12	Survey	CDC-National Immunization Survey Adult COVID Module (NIS-ACM)	Individual	329,135
3	Zhang et al. (2022) [18]	1	General Population	U.S.	18+	July 2021 to October 2021	Months 8–11	Survey	Household Pulse Survey (HPS)	Individual	321
4	Valier et al. (2023) [19]	1, 2, 3	General Population	U.S.	5–17	December 2020 to August 2022	Months 1–21	Survey	CDC-National Immunization Survey–Child COVID Module (NIS-CCM)	Individual	94,838
5	Elbel et al. (2023) [21]	2	Public Schools’ Students	New York City	5–18	May 2021 to May 2022	180 Days after FDA’s EUA for Each Children Age Group	Student Registry	NYC Public Schools’ Student Population Health Registry (SPHR)	Individual	1,077,311
6	Fast et al. (2022) [20]	3, 4	General Population	U.S.	5+	August 2022	Month 21	Immunization Registry	States and the D.C. immunization Registries’ Data Submitted to the CDC Immunization Information Systems	Individual	106,252,812

U.S: United States; D.C.: District of Columbia; CDC: The U.S. Center for Disease Control and Prevention; NYC: New York City; FDA: The U.S. Food and Drug Administration; EUA: Emergency Use Authorization.

**Table 3 healthcare-13-00139-t003:** The Included Articles’ Methods.


#	Authors (Year)	Analysis Type	Results Presentation	Demographic Multidimensionality	Age Groups	Sex Groups	Racial and Ethnic Groups	Quality Assessment
1	Diesel et al. (2021) [9]	Descriptive	Table at a Cross-Section of Time	Age–Sex	18–29	Female		Very Good
					30–49	Male		
					50–64			
					65+			
2	Kriss et al. (2022) [11]	Descriptive	Table at a Cross-Section of Time	Age–Ethnicity–Race	18–29		Non-Hispanic Black	Very Good
					30–49		Non-Hispanic White	
					50–64		Non-Hispanic Asian	
			Time Trends		65+		Non-Hispanic AI/AN	
							Non-Hispanic NH/OPI	
							Non-Hispanic Multiracial	
							Hispanic	
3	Zhang et al. (2022) [18]	Descriptive	Table at a Cross-Section of Time	Age–Ethnicity–Race	18–29	Female	Non-Hispanic Black	Good
				Sex–Ethnicity–Race	30–39	Male	Non-Hispanic White	
					40–49		Non-Hispanic Asian	
					50–59		Hispanic	
					60–69			
					70–79			
					80–88			
4	Valier et al. (2023) [19]	Descriptive	Table at a Cross-Section of Time	Age–Ethnicity–Race	5–11	Female	Non-Hispanic Black	Very Good
				Sex–Ethnicity–Race	12–15	Male	Non-Hispanic White	
				(Only for Dose One)	16–17		Non-Hispanic Asian	
			Time Trends				Non-Hispanic Other Races/Multiracial	
							Hispanic	
5	Elbel et al. (2023) [21]	Descriptive	Time Trends	Age–Ethnicity–Race	5–11		Non-Hispanic Black	Very Good
					12–15		Non-Hispanic White	
					16–18		Non-Hispanic Asian	
							Non-Hispanic Other Races/Multiracial	
							Hispanic	
6	Fast et al. (2022) [20]	Descriptive	Table at a Cross-Section of Time	Age–Ethnicity–Race	5–11	Female	Non-Hispanic Black	Very Good
				Age–Sex	12–17	Male	Non-Hispanic White	
					18–39		Non-Hispanic Asian	
					40–64		Non-Hispanic AI/AN	
					65+		Non-Hispanic NH/OPI	
							Non-Hispanic Other Races	
							Non-Hispanic Multiracial	
							Hispanic	

AI/AN: American Indian or Alaska Native; NH/OPI: Native Hawaiian or Other Pacific Islander.

## Data Availability

All records analyzed in this systematic review are publicly accessible and available.

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
