# Peer review of "Multidimensional Demographic Analyses of COVID-19 Vaccine Inequality in the United States: A Systematic Review"

_healthcare, 2025, doi:10.3390/healthcare13020139_

Round 1
Reviewer 1 Report
Comments and Suggestions for Authors
Thank you for the opportunity to review the manuscript titled „Multidimensional Demographic Analyses of COVID-19 Vaccine Inequality in the United States: A Systematic Review ".
The authors performed comprehensive demographic analyses of COVID-19 vaccine inequality in the United States. The authors conducted a systematic review concerning only the United States, therefore the article is more suitable for the national journal.
The manuscript is intriguing, and the authors have done commendable work. There are not many strong articles on this topic.
However, there are a few minor issues that the authors should address before the manuscript can be considered for publication.
Below are my comments detailing these issues.
1. Introduction
The authors do not address the topic of their work in the first paragraph (lines 32 to 41) or in the second paragraph (lines 42 to 52)and also in the third paragraph (lines 53 to 59), which might lead potential readers to feel that the title of the work is inadequate. It is only in the fourth paragraph (starting from line 60) that they finally get to the essence of their work.
The authors should consider revising the introduction to go straight to the main point of the article.
For authors and potential readers in the United States, the vaccination program against COVID-19 during the pandemic is well understood. However, for readers outside the United States, the distribution of the vaccine may not be as clear.
It is uncertain who received vaccinations first, whether based on age, race, profession, or other criteria. This can be explained in two or three sentences, allowing for comparison with systematic reviews from other countries.
3. Results
The numbers of documents taken into account contained in lines 179 to 188 do not match those described in and shown in Figure 1.
5. Conclusions
In conclusion, I kindly ask you to highlight the significance of your article more strongly, ( these elements were clearly articulated in the discussion).
Finally, please highlight the importance of your work.
Author Response
Comments and Suggestions for Authors
Thank you for the opportunity to review the manuscript titled „Multidimensional Demographic Analyses of COVID-19 Vaccine Inequality in the United States: A Systematic Review ".
The authors performed comprehensive demographic analyses of COVID-19 vaccine inequality in the United States. The authors conducted a systematic review concerning only the United States, therefore the article is more suitable for the national journal.
The manuscript is intriguing, and the authors have done commendable work. There are not many strong articles on this topic.
However, there are a few minor issues that the authors should address before the manuscript can be considered for publication.
Below are my comments detailing these issues.
- Introduction
The authors do not address the topic of their work in the first paragraph (lines 32 to 41) or in the second paragraph (lines 42 to 52)and also in the third paragraph (lines 53 to 59), which might lead potential readers to feel that the title of the work is inadequate. It is only in the fourth paragraph (starting from line 60) that they finally get to the essence of their work. The authors should consider revising the introduction to go straight to the main point of the article.
For authors and potential readers in the United States, the vaccination program against COVID-19 during the pandemic is well understood. However, for readers outside the United States, the distribution of the vaccine may not be as clear. It is uncertain who received vaccinations first, whether based on age, race, profession, or other criteria. This can be explained in two or three sentences, allowing for comparison with systematic reviews from other countries.
Authors’ Response: This makes perfect sense. Thank you for pointing it out. We have now added a paragraph to the beginning of the introduction section that directly introduces the study’s aim. It reads as:
“This study reviewed the evidence of inequalities in the COVID-19 vaccine uptake by combinations of three commonly reported demographic characteristics (age, sex, and ethnicity-race) in the United States (U.S.). The study focused on the U.S. to preserve homogeneity in the national immunization policies and the documentation of race and ethnicity.”
- Results
The numbers of documents taken into account contained in lines 179 to 188 do not match those described in and shown in Figure 1.
Authors’ Response: Thank you very much for finding the inconsistency. We have now corrected both the text and Figure 1.
- Conclusions
In conclusion, I kindly ask you to highlight the significance of your article more strongly, ( these elements were clearly articulated in the discussion).
Finally, please highlight the importance of your work.
Authors’ Response: Thank you for pointing it out. We have now added a sentence that highlights the significance and importance of our study at the beginning of the conclusion section. We also edited the next sentence accordingly. The updated first paragraph of the conclusion section is:
“This study is the first to systematically review peer-reviewed publications that analyzed multidimensional demographic evidence on COVID-19 vaccine uptake in the U.S. Such analyses often focused on the dose-one vaccination. Similar analyses of full vaccination (i.e., the receipt of the first two doses), especially in older adults, are rare but crucially important because of the marked age gradient of COVID-19 vulnerability.”

Reviewer 2 Report
Comments and Suggestions for Authors
A review of the manuscript entitled “Multidimensional Demographic Analyses of COVID-19 Vaccine Inequality in the United States: A Systematic Review”
1. (page 2, lines 53-55): This sentence “Age-specific COVID-19-related hospitalization rates illustrate similar nationwide trends as the COVID-19 mortality rates. The rate of COVID-19-related hospitalization was consistently higher among older individuals” will be more significant if it’s also supported by another relevant study. Please cite https://pubmed.ncbi.nlm.nih.gov/39280294/.
2. The Method should follow PRISMA guideline – please use subheading listed in PRISMA (Eligibility Criteria, Information Sources, Search Strategy, Study Selection, Data Collection Process, Data Items, Risk of Bias Assessment, Effect Measures, Synthesis Methods, Subgroup Analyses (if any), Reporting Bias Assessment).
3. In Results, please also follow the PRISMA subheading to ensure all aspects are included. See here - https://www.prisma-statement.org
4. In Figure 1. For those 210 studies, did authors read the full-text or not? If yes then please report the reason to excluded them.
5. The headings system used should be more accurate in the Results section
6. Where are the results of Joanna Briggs Institute (JBI)? Please provide.
7. More literature is required regarding implications of study findings
8. (page 14, lines 424-426): This sentence “The high rates of COVID-19 vaccination among non-Hispanic Asian individuals can be at least partially attributed to low vaccine hesitancy among them in comparison to those of other races and ethnicities” also shares the same idea with a study by Rosiello et al. https://pubmed.ncbi.nlm.nih.gov/38450212/. Kindly include this reference to make the sentence stronger.
9. Overall, this manuscript has presented a sophisticated review of multidimensional demographic inequalities in COVID-19 uptake. The use of good English has also been applied throughout the manuscript. However, the style of the English should be improved to be more scientific English.
Author Response
- (page 2, lines 53-55): This sentence “Age-specific COVID-19-related hospitalization rates illustrate similar nationwide trends as the COVID-19 mortality rates. The rate of COVID-19-related hospitalization was consistently higher among older individuals” will be more significant if it’s also supported by another relevant study. Please cite https://pubmed.ncbi.nlm.nih.gov/39280294/.
Authors’ Response: Thank you very much for providing the reference. We understand the importance of providing more evidence to support the statement. However, we took the liberty of referring to the following articles (all newly added) instead of the one suggested because of the U.S. focus of this study:
- Taylor CA, Patel K, Pham H, et al. COVID-19–Associated Hospitalizations Among U.S. Adults Aged ≥18 Years — COVID-NET, 12 States, October 2023–April 2024. MMWR Morb Mortal Wkly Rep 2024;73:869–875. DOI: http://dx.doi.org/10.15585/mmwr.mm7339a2
- Taylor CA, Patel K, Patton ME, et al. COVID-19–Associated Hospitalizations Among U.S. Adults Aged ≥65 Years — COVID-NET, 13 States, January–August 2023. MMWR Morb Mortal Wkly Rep 2023;72:1089–1094. DOI: http://dx.doi.org/10.15585/mmwr.mm7240a3
- Jean Y Ko, Melissa L Danielson, Machell Town, Gordana Derado, Kurt J Greenlund, Pam Daily Kirley, Nisha B Alden, Kimberly Yousey-Hindes, Evan J Anderson, Patricia A Ryan, Sue Kim, Ruth Lynfield, Salina M Torres, Grant R Barney, Nancy M Bennett, Melissa Sutton, H Keipp Talbot, Mary Hill, Aron J Hall, Alicia M Fry, Shikha Garg, Lindsay Kim, COVID-NET Surveillance Team, Risk Factors for Coronavirus Disease 2019 (COVID-19)–Associated Hospitalization: COVID-19–Associated Hospitalization Surveillance Network and Behavioral Risk Factor Surveillance System, Clinical Infectious Diseases, Volume 72, Issue 11, 1 June 2021, Pages e695–e703, https://doi.org/10.1093/cid/ciaa1419
- The Method should follow PRISMA guideline – please use subheading listed in PRISMA (Eligibility Criteria, Information Sources, Search Strategy, Study Selection, Data Collection Process, Data Items, Risk of Bias Assessment, Effect Measures, Synthesis Methods, Subgroup Analyses (if any), Reporting Bias Assessment).
Authors’ Response: Thank you very much for the comment. We have now restructured the methods section accordingly. We have added new notes and explanations as well.
- In Results, please also follow the PRISMA subheading to ensure all aspects are included. See here - https://www.prisma-statement.org
Authors’ Response: Thank you for another instructive comment. We have now reorganized the results section according to the PRISMA statement (Page et al. 2021: http://dx.doi.org/10.1136/bmj.n71). However, our subsections are not exactly the same as what the PRISMA statement suggests. They are: (3.1) Study Selection, (3.2) Study Characteristics: Population, Period, Data, and the Dose Numbers, (3.3) Study Characteristics: Methods, (3.4) Results. The main reasons for not statistically synthesizing the results were dissimilarities in the analyzed dose, study period, and categories of the subcategories of demographic characteristics. The risk of bias in the studies was reported within Section (3.3), at the end of the first paragraph.
- In Figure 1. For those 210 studies, did authors read the full-text or not? If yes then please report the reason to excluded them.
Authors’ Response: We read the full text of 217 articles and excluded 211 of them (we correcred the number of excluded articles to 211, as it was mistakenly reported 210 previously). We excluded the 211 articles because they did not meet the fourth (and the key) inclusion criterion, that is, they did not provide evidence on the COVID-19 vaccine uptake disparities in two or more combinations of age, sex, and ethnicity-race. In response to this comment, we replaced the following sentence in Section (3.1):
“Therefore, 217 studies were included for full-text review, which led to the exclusion of 211 documents that did not meet the inclusion criteria.”
with
“Therefore, 217 studies were included for full-text review, which led to the exclusion of 211 documents that did not meet a key inclusion criterion: providing evidence on the COVID-19 vaccine uptake disparities in two or more combinations of age, sex, and ethnicity-race.”
- The headings system used should be more accurate in the Results section
Authors’ Response: Thank you for the comment. We have now added a new subheading to the results section (3.1. The Review Process) and revised all numbered subheadings.
- Where are the results of Joanna Briggs Institute (JBI)? Please provide.
Authors’ Response: We used Munn et al. (2018) (https://doi.org/10.1186/s12874-018-0611-x) to justify that our study is a systematic review versus a scoping review, as we assessed the included articles risk of bias and synthesize their findings. We mistakenly mentioned that we were following the JBI guidelines. We have now corrected this error. We appreciate that you caught this error. In effect, we follow PRISMA guidelines and have now clarified it in response to your 2nd comment in the new Section 2.2.
- More literature is required regarding implications of study findings
Authors’ Response: Thank you very much for the comments. We have now added several new references to the articles. Also, we are referring to a number of newly added articles that investigate the effectiveness of COVID-19 vaccination in real-world data:
- Roberts EK, Gu T, Wagner AL, Mukherjee B, Fritsche LG. Estimating COVID-19 Vaccination and Booster Effectiveness Using Electronic Health Records From an Academic Medical Center in Michigan. AJPM Focus. 2022 Sep;1(1):100015. doi: 10.1016/j.focus.2022.100015.
- Surie D, DeCuir J, Zhu Y, et al. Early Estimates of Bivalent mRNA Vaccine Effectiveness in Preventing COVID-19–Associated Hospitalization Among Immunocompetent Adults Aged ≥65 Years — IVY Network, 18 States, September 8–November 30, 2022. MMWR Morb Mortal Wkly Rep 2022;71:1625–1630. DOI: http://dx.doi.org/10.15585/mmwr.mm715152e2
- artof SY, Slezak JM, Puzniak L, et al. Effectiveness of BNT162b2 BA.4/5 bivalent mRNA vaccine against a range of COVID-19 outcomes in a large health system in the USA: a test-negative case-control study [published correction appears in Lancet Respir Med. 2023 Dec;11(12):e98. doi: 10.1016/S2213-2600(23)00422-8]. Lancet Respir Med. 2023;11(12):1089-1100. doi:10.1016/S2213-2600(23)00306-5
The references indicate a general policy implication of the study, the importance of COVID-19 vaccination.
- (page 14, lines 424-426): This sentence “The high rates of COVID-19 vaccination among non-Hispanic Asian individuals can be at least partially attributed to low vaccine hesitancy among them in comparison to those of other races and ethnicities” also shares the same idea with a study by Rosiello et al. https://pubmed.ncbi.nlm.nih.gov/38450212/. Kindly include this reference to make the sentence stronger.
Authors’ Response: Thank you very much for providing the reference. If it is okay, given the U.S. focus of this study, we cited the following articles (all newly added):
- Don E. Willis, Brooke E.E. Montgomery, James P. Selig, Jennifer A. Andersen, Sumit K. Shah, Ji Li, Sharon Reece, Derek Alik, Pearl A. McElfish, COVID-19 vaccine hesitancy and racial discrimination among US adults, Preventive Medicine Reports, Volume 31, 2023, https://doi.org/10.1016/j.pmedr.2022.102074.
- Ratnayake, A., Hernandez, J.H., Justman, J. et al. Vaccine Hesitancy at Nine Community Sites Across the United States, Early in COVID-19 Vaccine Rollout. J. Racial and Ethnic Health Disparities (2024). https://doi.org/10.1007/s40615-024-02172-0
- Overall, this manuscript has presented a sophisticated review of multidimensional demographic inequalities in COVID-19 uptake. The use of good English has also been applied throughout the manuscript. However, the style of the English should be improved to be more scientific English.
Authors’ Response: Thank you very much for your careful review of the article and very helpful comments. In addition to addressing your and other reviewers’ comments, we have reread and revised the articles several times and worked on the style. For example, we made sure that a passive tone was consistently used throughout the document, added clarifications to the introduction, methods, and results sections wherever needed, reviewed the use of dash signs, used font bolding consistently, and shortened some long sentences.

Reviewer 3 Report
Comments and Suggestions for Authors
Thank you for the opportunity to review this manuscript. The authors present a systematic review of studies of COVID-19 vaccination uptake across demographic groups in the US. The authors identified 6 eligible records from 2,793 initial records. This systematic review builds on existing research into inequalities in vaccine uptake, which have previously only focused on inequalities across one demographic characteristic (e.g., age, sex, ethnicity/race), by examining multidimensional inequalities. However, this review is limited in that it solely focuses on the US context, despite several regions from around the world showing similar inequalities in COVID-19 health outcomes, and the justification for focusing on the US is not clear. In addition, the methodological approach to data synthesis and analysis is not outlined, and as a consequence of this, the results section is limited to a description of each study. The authors then summarise consistent patterns from across the eligible studies in the discussion, but this would be more suitable in the results, enabling a more in-depth evaluation of the importance of these findings in the discussion. I have expanded on these points below:
Abstract:
1. Be clear in the abstract that this review focusses solely on evidence from the US.
Introduction:
1. The rationale for focussing on the US is unclear, when many other regions experienced similar inequalities in COVID-19 health outcomes and vaccine uptake. This should be justified in the first paragraph, as I was initially confused by the focus on US literature only. The limitations of this should be outlined in the discussion.
Materials and methods:
1. Given that this review refers to COVID-19 vaccine uptake and the literature has evolved rapidly, it may be worth updating the search, as the initial search was conducted a year ago.
2. I am surprised at the relatively low number of records that were identified in the search and because the full search strategy is not reported, it’s unclear whether records had to include all of the key concepts (in the full text, abstract, title, key words?). It may be useful to include the full search strategy for one database in supplementary materials.
3. What was the rationale for focusing only on age, sex, and ethnicity-race, rather than expanding to include other demographic characteristics, e.g., socioeconomic position, marital status?
4. Further detail is needed regarding the eligibility criteria, as the article later states that among the excluded documents, “one examined multidimensional COVID-19 vaccine uptake only through regression analyses” and it’s not clear why it was excluded.
5. The methods should include a section on data synthesis and analysis. If meta-analyses were not possible, what methods were used to synthesise the findings?
6. Consider including a GRADE approach to strengthen this review.
Results/Discussion:
1. Some aspects of the PRISMA flow chart are completed incorrectly – the reports assessed for eligibility should be n = 217 and then the reports excluded should be 212, stratified by reasons for exclusion. The reports not retrieved should be the number of articles for which you could not find the full text.
2. Reference is needed for the PRISMA flow chart in Figure 1 https://www.bmj.com/content/372/bmj.n71
3. The results are difficult to follow, which is in part due to the methodological approach to data synthesis not being outlined. The authors describe findings of each individual study, rather than synthesising what the findings mean, taken together
4. In the discussion, the authors summarise consistent patterns from across the included studies. However, this is what should be reported in the results section. The discussion should then focus on interpreting the findings, in line with wider literature, and outlining the implications of these findings. Both the results and discussion are very descriptive with little analysis.
Author Response
Thank you for the opportunity to review this manuscript. The authors present a systematic review of studies of COVID-19 vaccination uptake across demographic groups in the US. The authors identified 6 eligible records from 2,793 initial records. This systematic review builds on existing research into inequalities in vaccine uptake, which have previously only focused on inequalities across one demographic characteristic (e.g., age, sex, ethnicity/race), by examining multidimensional inequalities.
However, this review is limited in that it solely focuses on the US context, despite several regions from around the world showing similar inequalities in COVID-19 health outcomes, and the justification for focusing on the US is not clear. In addition, the methodological approach to data synthesis and analysis is not outlined, and as a consequence of this, the results section is limited to a description of each study. The authors then summarise consistent patterns from across the eligible studies in the discussion, but this would be more suitable in the results, enabling a more in-depth evaluation of the importance of these findings in the discussion. I have expanded on these points below:
Abstract:
- Be clear in the abstract that this review focusses solely on evidence from the US.
Authors’ Response: The clarification is important. Thank you for pointing it out. We have now specified that the study focuses on the U.S. in both background and objective sections of the abstract.
Introduction:
- The rationale for focussing on the US is unclear, when many other regions experienced similar inequalities in COVID-19 health outcomes and vaccine uptake. This should be justified in the first paragraph, as I was initially confused by the focus on US literature only. The limitations of this should be outlined in the discussion.
Authors’ Response: We understand the importance of justifying the U.S. focus of this study. We have now added a paragraph to the beginning of the introduction section that directly introduces the study’s aim and geographical focus and the reason for the latter. It reads as:
“This study reviewed evidence of inequalities in the COVID-19 vaccine uptake by combinations of three commonly reported demographic characteristics (age, sex, and ethnicity-race) in the United States (U.S.). The study focused on the U.S. to preserve homogeneity in the national immunization policies and the documentation of race and ethnicity.”
Also, as suggested, we have listed this among the limitations of the study (as the first limitation) in the last paragraph of the discussion section:
“First, it solely focuses on the U.S. Hence, its results are not generalizable to other regions and countries.”
Materials and methods:
- Given that this review refers to COVID-19 vaccine uptake and the literature has evolved rapidly, it may be worth updating the search, as the initial search was conducted a year ago.
Authors’ Response: This study has been underway for more than 2 years and has gone through numerous updates in aims and scope, hence keywords and search strings changes. In the current scope and aims, we updated the search once in January and February 2023. Then, the review took about 6 months to be completed due to the number of articles found (mainly because unidimensional demographic evidence on COVID-19 vaccine inequality is abundant) and the level of review precision needed to identify multidimensional evidence. Another major challenge was synthesizing the results of the included studies, as most of them used different age groups and were not focused on a specific vaccine dose. These all contributed to the amount of time needed for the manuscript to be completed. In our assessment, a search update will add about 6 months for a manuscript update, while we believe the search period (1/1/2020 to 12/31/2023) captures almost all the evidence on the topic, mainly due to the significant decrease in COVID-19-related research. Therefore, we respectfully request that we stay within the current time range. Nonetheless, we have listed this among the limitations of the study (as the second limitation) in the last paragraph of the discussion section:
“Second, the study covers the first three years (2021-2023) of the COVID-19 vaccination campaign in the U.S.; hence, it is possible that more multidimensional evidence on COVID-19 vaccine uptake was collected in 2024.”
- I am surprised at the relatively low number of records that were identified in the search and because the full search strategy is not reported, it’s unclear whether records had to include all of the key concepts (in the full text, abstract, title, key words?). It may be useful to include the full search strategy for one database in supplementary materials.
Authors’ Response: We were also surprised by the low number of included records. The low number indicates how scarce multidimensional evidence is on the uptake of the COVID-19 vaccine in the U.S. We have now provided our search strategy in a methods appendix, updated the methods section, and provided more details in response to this comment.
- What was the rationale for focusing only on age, sex, and ethnicity-race, rather than expanding to include other demographic characteristics, e.g., socioeconomic position, marital status?
Authors’ Response: It is certainly essential to provide the rationale. We have now revised the first paragraph of the materials and methods section to address it. The newly added segment is:
“These demographic characteristics are commonly reported with minimal variations, especially in race and ethnicity, in the U.S. data such as decennial censuses, immunization registries, and population, community, and household surveys. However, socioeconomic characteristics such as education, marital status, and family income are not commonly reported in U.S. data. Particularly, U.S. immunization registries do not contain such information.”
In addition, we added another limitation to the list of the study’s limitations that points out to this matter:
“Third, this study did not review multidimensional evidence on COVID-19 vaccine inequality by socioeconomic characteristics (e.g., education, marital status, and income), mainly because such information is not reported in U.S. immunization registries and at the individual level in census data. Therefore, reviewing the analyses of the intersection of demographic and socioeconomic factors remains an area of development in the COVID-19 vaccination literature.”
- Further detail is needed regarding the eligibility criteria, as the article later states that among the excluded documents, “one examined multidimensional COVID-19 vaccine uptake only through regression analyses” and it’s not clear why it was excluded.
Authors’ Response: The reason for our focus on descriptive evidence was to provide direct evidence on multidimensional COVID-19 vaccine uptake inequality. Regression analyses, on the other hand, provide indirect evidence of inequality as they adjust the estimates of demographic inequalities with aggregate and individual-level covariates such as poverty rate, income, education, and political party affiliation. To clarify this, we have now updated the sentence (which is located in the first paragraph of the results section) to:
“Among the 6 included documents, one that examined multidimensional COVID-19 vac-cine uptake only through regression analyses was excluded as it provided indirect, co-variate-adjusted estimates of multidimensional COVID-19 vaccine uptake inequality.”
- The methods should include a section on data synthesis and analysis. If meta-analyses were not possible, what methods were used to synthesise the findings?
Authors’ Response: Thank you for the questions. We have now reorganized the methods section and added more explanations. The new section 2.8 discusses this matter now.
- Consider including a GRADE approach to strengthen this review.
Authors’ Response: We appreciate the suggestion. We actually considered using the GRADE approach for the assessment of the included studies but found the Standard Quality Assessment Criteria (Kmet et al. 2004: https://doi.org/10.7939/R37M04F16) more appropriate for this study. The GRADE approach evaluates imprecision, inconsistency, indirectness, and publication bias in addition to the risk of bias in a study.
We did not adopt a GRADE approach because (1) we are not evaluating the effect of a specific clinical, medical, or policy intervention in this study, (2) the effect measure (i.e., the rate of vaccination) and the key analysis factors (i.e., the four demographic characteristics: age, sex, race, and ethnicity) are simple and reported with rather minimal imprecision or inconsistency, and (3) the analyses we included were all descriptive.
Instead, we found the Standard Quality Assessment Criteria more straightforward and to the point in this study. In this method, we answered the following questions for each included study:
(1) Is the question / objective sufficiently described?
(2) Study design evident and appropriate?
(3) Method of subject/comparison group selection or source of information/input variables described and appropriate?
(4) Subject (and comparison group, if applicable) characteristics sufficiently described?
(5) If interventional and random allocation was possible, was it described?
(6) If interventional and blinding of investigators was possible, was it reported?
(7) If interventional and blinding of subjects was possible, was it reported?
(8) Outcome and (if applicable) exposure measure(s) well defined and robust to measurement / misclassification bias? Means of assessment reported?
(9) Sample size appropriate?
(10) Analytic methods described/justified and appropriate?
(11) Some estimate of variance is reported for the main results?
(12) Controlled for confounding?
(13) Results reported in sufficient detail?
(14) Conclusions supported by the results?
We found answering 11 out of the 14 questions easily answerable for the included studies; only questions (5), (6), and (7) were not applicable to our study.
Results/Discussion:
- Some aspects of the PRISMA flow chart are completed incorrectly – the reports assessed for eligibility should be n = 217 and then the reports excluded should be 212, stratified by reasons for exclusion. The reports not retrieved should be the number of articles for which you could not find the full text.
Authors’ Response: Thank you for pointing out the error. We have now corrected it.
- Reference is needed for the PRISMA flow chart in Figure 1 https://www.bmj.com/content/372/bmj.n71
Authors’ Response: Included. Thank you.
- The results are difficult to follow, which is in part due to the methodological approach to data synthesis not being outlined. The authors describe findings of each individual study, rather than synthesising what the findings mean, taken together
Authors’ Response: Since there are key commonalities among the studies, we think it is important to synthesize the findings of the included studies to some degree. The commonalities are in the studies' dose number, large age group (children vs. adults vs older adults), and demographic multidimensionality. Therefore, we organized the description of the results by large age group within the demographic combination and then within the dose number. Specifying the three characteristics (dose number, multidimensionality, and large age groups) has policy implications as well. To make our points clearer, we have now revised the organization of the methods and results sections, added or edited subheadings, and included more explanations. We hope these changes helped with clarification.
- In the discussion, the authors summarise consistent patterns from across the included studies. However, this is what should be reported in the results section. The discussion should then focus on interpreting the findings, in line with wider literature, and outlining the implications of these findings. Both the results and discussion are very descriptive with little analysis.
Authors’ Response: Thank you very much for the comment. The descriptive nature of our results section was intentional, as we wanted to synthesize the included studies’ findings as they were. As the results section is already too long, we reviewed the key findings of the studies in the discussion section. We tried designing the discussion section in several different ways. For example, we thought of creating four separate subsections in the discussion: (1) a summary of important study characteristics, (2) key findings of the studies, (3) relevance of the key findings to the literature, (4) policy implications, and (5) limitations of our study. Our interpretations of the results would be more apparent if we had chosen that structure, but we figured the discussion would flow more naturally if we combined the subsections (2), (3), and (4). As such, the first three paragraphs of the discussion section were assigned to providing a summary of important study characteristics. Then, we discussed each key finding in terms of its relevance to the literature and policy implications in a separate paragraph. We hope this explanation clarifies our strategy. We have also created subsections within the discussion section. If you think they are insufficient, we can certainly revise the section more.
